

1       Particle size distribution and new particle formation under influence of biomass

2       burning at a high altitude background site of Mt. Yulong (3410m) in China

Dongjie Shang[1], Min Hu[1,2]\*, Jing Zheng[1], Yanhong Qin[1], Zhuofei Du[1], Mengren Li[1],

Jingyao Fang[1], Jianfei Peng[1], Yusheng Wu[1], Sihua Lu[1], Song Guo[1]\*

[1]State Key Joint Laboratory of Environmental Simulation and Pollution Control,

College of Environmental Sciences and Engineering, Peking University, Beijing,

100871, China

[2]Beijing Innovation Center for Engineering Sciences and Advanced Technology,

Peking University, 100871, Beijing, China

\* Corresponding author: E-mail address: minhu@pku.edu.cn, guosong@pku.edu.cn

**Abstract**

Biomass burning (BB) activities have a great impact on particle number size

distribution (PNSD) in upper troposphere of Tibet-Plateau, which could affect

regional and global climate. The intensive campaign for the measurement of PNSD,

gaseous pollutants and meteorological parameters was conducted at Mt. Yulong, a

high-altitude site (3410 m a. s. l.) in the southeast of Tibet Plateau during the

pre-monsoon season (22 March to 15 April), when the intensive BB activities in South

Asia were observed by fire maps. Long-range transport of BB pollutants could

increase the accumulation mode particles in background atmosphere of Mt. Yulong.

As a consequence, cloud condensation nuclei (CCN) concentration was found to be

2-8 times higher during BB periods than that during clean period. Apart from BB,

variation of planet boundary layer (PBL) and new particle formation were other

factors that influenced PNSD. However, only 3 NPF events (with a frequency of 14 %)

were observed at Mt. Yulong. Occurrence of NPF events during clean episode

corresponded with elevated PBL or transported BB pollutants. Due to lack of

condensable vapors including sulfuric acid and organic compounds, the newly formed

particles were not able to grow to CCN size. Our study emphasized the influences of

BB on aerosol and CCN concentration in atmosphere of Tibet Plateau. These results



can improve our understanding of the variation of particle concentration in upper
troposphere, and provide information for regional and global climate models.
Key words: Tibet-Plateau, particle number size distribution, biomass burning,
CCN, new particle formation

**1.** Introduction

The aerosol particles can influence the radiation of the planet surface through
scattering the sunlight, and cloud albedo by serving as cloud condensation nuclei
(CCN) (IPCC, 2013). The cloud albedo effect of aerosols provides the biggest
uncertainties in global climate models (IPCC, 2013), and depends strongly on number
concentration and size of particles. Numerous studies concentrated on monitoring
particle number size distribution (PNSD) within the planet boundary layer (PBL),
where anthropogenic sources have strong impacts (Peng et al., 2014). While the
particles in pristine free troposphere (FT) were rarely studied. Particles in FT mainly
originated from lifting of emission within PBL by convective, frontal, and orographic
lifting (Okamoto and Tanimoto, 2016), or atmospheric nucleation. Those particles
have longer lifetime and could be transported in a longer distance, during which they
could exchange with PBL (Shen et al., 2016;D'Andrea et al., 2016). Hence, studies on
FT are important because : 1) CCN in FT could influence cloud albedo more directly
compared to surface CCN; 2) FT served as a route of long-range transport of
pollutants. Aircraft study is a direct way to measure the FT particles, but it is costly
and can only provide data within short periods. Therefore, measurement at high
mountain sites is one common method to study the FT particles and analyze the
influences of pollution transport on FT (Shen et al., 2016).
Particles originated from BB in South Asia could have impacts on vast
atmosphere of Tibet Plateau by transport in FT. As the highest plateau in the world,
Tibet Plateau has very few anthropogenic sources, and could be taken as the continent
background. However, recent studies revealed that the smoke plume in South Asia
could ascend to FT, and transport to Himalayas and the mountain valley of Tibet
Plateau during pre-monsoon season (Cong et al., 2015;Lüthi et al., 2015;Bukowiecki
et al., 2016). During pre-monsoon season, the enhanced convection and steep pressure





gradient across the Himalaya-Gangetic region could rise the BB particles to higher
altitude (Gautam et al., 2009;Adak, 2014). The particles could be transported by dry
westerly, and have impacts on aerosols in Tibet Plateau region (Bonasoni et al.,
2010;Chen et al., 2014).   Former studies verify South Asian BB's influence in Tibet
Plateau by chemical analysis of $K^+$, levoglucosan, etc. However, there was limited
information of variation of PNSD under influence of BB. Also, there were limited
studies concerning the contribution of BB to CCN in Tibet Plateau.

Except primary emissions, new particle formation (NPF) is another important

source of particles in FT, but with limited measurement. According to model results,
nucleation in FT contribute to 35 % of the CCN globally (Merikanto et al., 2009).
Considering the level of pre-existing particles in FT is relatively low, it should
provide a good condition for nucleation of the nanoparticles. As a result, NPF has
been observed to happen frequently in FT, including Mt. Tai (1500m a.s.l.) (Shen et
al., 2016), Mediterranean Sea (1000m-300m a.s.l.) (Rose et al., 2015), Mt. Puy de
Dôme (1465 m a.s.l), Mt. Izana (2367m a.s.l.) (Rodríguez et al., 2009;García et al.,
2014), Colorado Rocky Mountains (2900m a.s.l.) (Boy et al., 2008), etc. While NPF
events happened less frequently at Indian foothill Himalayas (2080m) (Neitola et al.,
2011). Studies at mountain sites considered that the frequency of NPF corresponded
to the rise of PBL height, which could raise the concentration of anthropogenic $SO_2$,
$NH_3$ and other nucleation precursors. Mechanisms of formation and the growth of
nanoparticles in FT remain ambiguous (Bianchi et al., 2016), thus comprehensive
measurements of PNSD as well as trace gases at high-mountain sites are necessary to
provide information around this topic.

This study aimed to: 1) investigate the influence of BB from South Asia on

PNSD and CCN concentration at South east of Tibet Plateau; 2) characterize the NPF
at high-mountain sites. For purposes of these, a comprehensive measurement was
conducted at a background site in Mt. Yulong (3140 m a.s.l.), during the pre-monsoon
season.
**2.** Experiments and data analysis



## 2.1 Monitoring site

An intensive field campaign was conducted during 22 March to 15 April, at a high mountains site of Mt. Yulong (27.2N, 100.2E) in Southwest China and Southeast corner of Tibet Plateau, with an altitude of 3140 m a. s. l. This site is one of national regional background sites coordinated by the Chinese Environmental Monitoring Center (CEMC), which is a remote site on the transport route of South Asian pollutants during pre-monsoon season. At the foot of the Mt. Yulong, 36 km to the south of the site is the famous Lijiang Old Town, a populated tourist place. More details of the monitoring site can be found in another paper (Zheng et al., 2017).

## 2.2 Instrumentation

PNSD was measured with a time resolution of 5 min, by two set of scanning mobility particle sizer (SMPS, TSI Inc., St. Paul, MN, USA) and an aerodynamic particle sizer (APS, TSI model 3321, TSI Inc., St. Paul, MN, USA). The first set of SMPS consisted of a short differential mobility analyzer (DMA, Model 3085) and an ultra-condensing particle counter (UCPC, Model 3776, flowrate 1.5 L/min) was used to measure the 3-60 nm particles. Another SMPS with long DMA (Model 3081) and normal CPC (Model 3022, flow rate 0.3 L/min) was used for measuring 60-700 nm particles. A silicon diffusion tube was placed before the SMPS, controlling the relative humidity of sampling air under 35 %. Diffusion loss and multiple charging calibration of the particles was done for SMPS data. APS with flow rate of 1 L/min was used for measuring 0.5-10 μm particles. The result of APS was modified to stokes diameter assuming the particle density to be 1.7 $\mu g/m^3$ before combining with SMPS data. A bypass flow was added before the inlet cutoff, to meet the working flow rate of the $PM_{10}$ cyclone (16.7 L/min).

To investigate the BB influences in aerosols, a high-resolution time-of-flight aerosol mass spectrometer (HR-TOF-AMS) was deployed to measure the chemical composition of aerosols. Through this instrument, we can obtain the concentration of nitrate, sulfate, ammonium, chloride and high-resolution mass spectrum of organics,



especially the fragments of BB organic markers. Black carbon (BC) is another
important marker for combustion sources. In this study, BC was measured with an
aethalometer (Magee Scientific, USA, type AE31), by collecting aerosol particles on a
filter stripe, and analyzing the transmission of the lights with seven wave length, from
370 to 950 nm. BC concentration was calculated as a multiple of the light absorption
coefficient at 880nm, with the default mass attenuation cross sections of 16.6 $m^2\ g^{-1}$
(Fröhlich et al., 2015). To get the concentration of organic tracers of the new particle
formation, an online-gas chromatography coupled with mass spectrometer and flame
ionization detectors (GC-MS/FID) was used to measure the non-methane
hydrocarbons (NMHCs), including benzene, toluene, monoterpene, etc.
Meteorological parameters, $PM_{2.5}$ and trace gases were also measured by online
instruments during the campaign (Table S1).

2.3 Data processing
2.3.1 Backward trajectory analysis
The 48h backward trajectories of the air mass were computed at 4000 m a.s.l.
(600 m above the ground of the Mt. Yulong site) by the Weather Research and
Forecasting (WRF) model (version 3.61) to identify the impacts from South Asia. The
fire spots were obtained from the satellite map from Moderate Resolution Imaging
Spectroradiometer (MODIS) (https://firms.modaps.eosdis.nasa.gov/firemap/). In order
to characterize the air mass origin during the NPF events, the 48h backward
trajectories at 600 m above the ground were calculated by NOAA HYSPLIT 4
(Hybrid Single-Particle Lagrangian Integrated Trajectory) model (Draxier and Hess,

1998).

2.3.2 Parameterization of NPF
The data of each PNSD during NPF was fitted as the sum of three or two mode
lognormal distribution (Hussein et al., 2005), including the geometric mean diameter



$D_m$, geometric standard deviation $\sigma_m$ and total number concentration of each mode.
During the NPF events, the growth rate (GR) was calculated as the variation of the
mean diameter $D_m$ of newly formed mode in unit internal:
$$GR = \frac{\Delta D_m}{\Delta t} \tag{1}$$

Formation rate was calculated for nucleation fraction of the particles (3-25 nm), with
the formula:
$$J_{3-25} = \frac{dN_{3-25}}{dt} + N_{3-25} \cdot CoagS_8 + F_{growth} \tag{2}$$

in this formula, $N_{3-25}$ is number concentration of particles within size range of 3-25
nm, $CoagS_8$ is the coagulation rate of particles with diameter of 8 nm, which is the
geometric mean of 3-25 nm. The coagulation rate was calculated as:
$$CoagS(D_p) = \int K(D_p, D_p') n(D_p') dD_p' \tag{3}$$

in which $n(D_p')$ is number concentration of particles with size of $D_p'$,
$K(D_p, D_p')$ is the coagulation coefficient between $D_p$ and $D_p'$ particles. During
nucleation events, there were negligible particles that grew beyond 25 nm, so the last
term in formula of was not included (Dal Maso et al., 2005). To quantify the
limitation of NPF from pre-existing particles, the condensation sink was calculated as:
$$CS = 2\pi D \sum_i \beta \cdot D_i \cdot N_i \tag{4}$$

where D is the diffusion coefficient of the condensational vapor, e.g. sulfuric acid, $\beta$
the transitional regime correction factor, $D_i$ and $N_i$ are the diameter and number
concentration of particles in class i. In calculation described above, all diameters were
dry diameter directly measured from SMPS, so the coagulation and condensation sink
could be underestimated.
Sulfuric acid was thought to be the most important precursor of NPF events
(Sipilä et al., 2010), and could contribute to particle growth by condensation (Yue et
al., 2010;Zhang et al., 2012). In this study, the content of $H_2SO_4$ was calculated by a
pseudo-steady state method (Kulmala et al., 2001):
$$[H_2SO_4] = k \cdot [OH][SO_2]/CS \tag{5}$$

in which [OH] and [$SO_2$] are number concentration of OH radicals and $SO_2$, value of



k is $10^{-12}$ cm$^3$s$^{-1}$. [OH] was estimated by:
$$[OH] = a(JO^1D)^\alpha (J_{NO2})^\beta \frac{b[NO_2]+1}{c[NO_2]^2+d[NO_2]+1} \tag{6}$$

in which α=0.83, β=0.19, a=4.1×10$^9$, b=140, c=0.41, d=1.7 (Ehhalt and Rohrer,
2000). Contribution of sulfuric acid condensation to particle growth was calculated by
Yue et al 's (2010) method.
2.3.3 Calculation of CCN concentration

In order to evaluate the variation of indirect climate effects of the particles at Mt.

Yulong, CCN number concentration was estimated from data of PNSD and particle
chemical composition. Firstly, the SNA (sulfate, nitrate, ammonium) was ion-coupled
to get exact chemical compounds of the inorganic salts in particles. NH$_4$NO$_3$, H$_2$SO$_4$,
NH$_4$HSO$_4$ and (NH$_4$)$_2$SO$_4$ were calculated following the formula:
$$n_{\text{NH}_4\text{NO}_3} = n_{\text{NO}_3^-},$$

$$n_{\text{H}_2\text{SO}_4} = \max(0,\ n_{\text{SO}_4^{2-}} - n_{\text{NH}_4^+} + n_{\text{NO}_3^-}),$$

$$n_{\text{NH}_4\text{HSO}_4} = \min(2n_{\text{SO}_4^{2-}} - n_{\text{NH}_4^+} + n_{\text{NO}_3^-},\ n_{\text{NH}_4^+} - n_{\text{NO}_3^-}),$$

$$n_{(\text{NH}_4)_2\text{HSO}_4} = \max(n_{\text{NH}_4^+} - n_{\text{NO}_3^-} - n_{\text{SO}_4^{2-}},\ 0),$$

where $n$ is the mole number of the specific compounds (Gysel et al., 2007). Based on
κ-Köhler theory and Zdanovskii–Stokes–Robinson (ZSR) mixing rule, the
hygroscopic parameter of mixed particles can be calculated as (Petters and
Kreidenweis, 2007):
$$\kappa = \sum_1^n \varepsilon_m \kappa_m$$

where $\varepsilon_m$ is the volume fraction of the composition $m$ in particles, and $\kappa_m$ is the
hygroscopic parameter of pure composition $m$. In this research, we consider
secondary inorganic ions, organics and BC as majority composition of particles, and
put them into the ZSR mixing formula. The correlated parameters of the compounds
we used are in table 1.
**Table 1. Densities and hygroscopic parameters of the compounds used in CCN**
**calculation**





| Species | NH$_4$NO$_3$ | NH$_4$HSO$_4$ | (NH$_4$)$_2$SO$_4$ | H$_2$SO$_4$ | Organics | BC |
|---|---|---|---|---|---|---|
| ρ (kg m$^{-3}$) | 1720 | 1780 | 1769 | 1830 | 1400 | 1700 |
| κ | 0.67 | 0.61 | 0.61 | 0.91 | 0.1 | 0 |


Based on κ-Köhler theory, the relationship between κ and D$_c$ under certain
supersaturation (S$_c$) is:
$$\kappa = \frac{4A^3}{27D_c^3 \ln^2 S_c}, \quad A = \frac{4\sigma_{s/a}M_w}{RT\rho_W}$$

in which $\sigma_{s/a}$ is the surface tension of water, M$_w$ and $\rho_W$ is the molecular weight and
density of water respectively, R is 8.317 J · mol$^{-1}$ · K$^{-1}$, T is the ambient temperature.
With the κ of the particles, the critical diameter D$_c$ of the CCN activation can be
achieved with this formula. Then the number concentration of CCN can be calculated
as number concentration of particles larger than D$_c$.
**3.**  Results and discussion
3.1 Particle number size distribution
3.1.1 Particle and meteorology parameters
Fig.1 shows the time series of PNSD and correlating meteorological parameters.
Temperature and relative humidity was 6.1±3.5 °C and 54.9±19.7 %, respectively
(Fig.1b). Southeast wind was dominant during the campaign, followed by South wind
and Southwest wind. Average wind speed was 2.9±1.8 m/s (Fig S1). Most of the
monitoring days were sunny, in favor of nucleation process, while short time rainfall
occurred on 24, 26 March and 4, 6, 7, 8, 10, 11 April. During April 12, there was a
heavy snow with the RH more than 90 %.
As a background high altitude site in TP, Mt. Yulong site revealed the feature of
low particle concentration and strong oxidation capacity. On average PM$_{2.5}$ was
10.51±9.16 μg/m$^3$, similar with the result on Northeast slope of Tibet Plateau (Xu et
al., 2014). This result was only 1/10-1/3 of that in the atmosphere of urban and rural
regions in China, indicating a background situation in Southwest China (Zheng et al.,



2016). However, the PM$_{2.5}$ at Yulong background site during the monsoon season was
around 3 times as that at a Qilian Shan Station in Northeast of Tibet Plateau (Xu et al.,
2015) and at Jungfraujoch, Switzerland (Bukowiecki et al., 2016), with similar
altitude, indicating relatively stronger anthropogenic influence in Southeast Tibet
Plateau. During 22 to 30 March, 4 to 5 April and 11 to 12 April, particle mass
concentration exceeded 10 μg/m$^3$, building up a pollution episode.

During the measurement, ozone level was 50.1±7.0 ppbv, similar with the results

at high mountain sites in Europe (Cristofanelli et al., 2016;Okamoto and Tanimoto,
2016), higher than the results in Beijing during spring. The concentration of NO$_x$ and
NO was 0.94±0.62 ppbv and 0.07±0.05 ppbv, respectively. SO$_2$ concentration was
0.06±0.05 ppbv, around the detection limit, showing no strong primary pollution. CO
concentration was 0.22±0.07ppmv, and showed higher level during the start of the
campaign (24 to 30 March), which could be resulted from the influence of BB (Fig
1d).

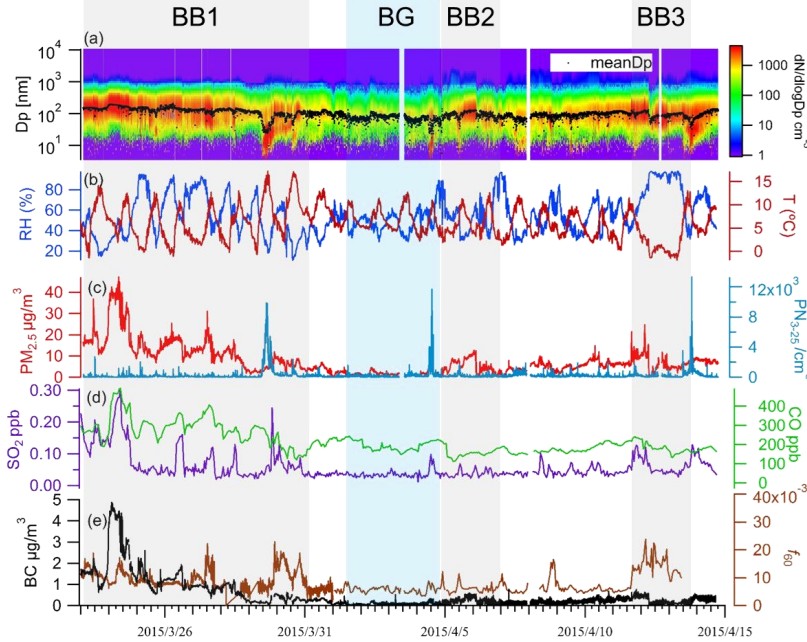


**Figure 1. Time series of (a) particle number size distribution and geometric mean**
**diameter, (b) ambient temperature, relative humidity, (c) PM$_{2.5}$ mass**



**concentration, number concentration of nucleation mode (3-25 nm) particles**
**(PN$_{3-25}$), (d) SO$_2$ and CO concentration, (e) black carbon concentration, fraction**
**of $f$60 (organic fragment ions with m/z=60) during the monitoring campaign.**
**Periods influenced by biomass burning (BB1, BB2, BB3) were marked by grey**
**shades, period representing background condition (BG) was marked by blue**
**shade.**
Although particles we measured in this study had larger size range than most of
other studies, the results can still be comparable, considering that Aitken and
accumulation mode particles, which all measurements included, constitute most of the
particle number concentration (PN). Table 2 showed particle number concentrations in
atmosphere at Mt. Yulong and other high altitude stations. Total number concentration
of PM$_{10}$ was 1600$\pm$1290 cm$^{-3}$ during monsoon season of Mt. Yulong, slightly lower
than those measured at other sites around Tibet Plateau, e.g. Waliguan and
Mukteshwar, and Mt. Huang. However, this result is several times higher than those
of areas with scarce emission sources, e.g. Alps and Antarctica. On the other hand, PN
didn't show clear trend as the altitude increases, which means the regional emission
and transport had larger impact on aerosols in upper troposphere, rather than the
vertical distribution. We define N$_{3-25}$, N$_{25-100}$, N$_{100-1000}$, N$_{1000+}$ as number
concentrations of particles with diameters of 3-25 nm, 25-100 nm, 100-1000 nm and
1-10 μm, respectively. There were bursts of N$_{3-25}$ on midday of 29 March, 4 April, 13
April, with the peak value at 9900 cm$^{-3}$, 11700 cm$^{-3}$ and 5400 cm$^{-3}$, respectively
(Fig.1c). During those periods, the geometric mean diameter of the particles was
lower than 25 nm. Those events could be resulted from local or regional new particle
formation, which would be discussed later.









**Table 2. Particle number concentration of high altitude sites around the world, in**
**comparison with this study**

| Location | Altitude [m] | Date | Size range [nm] | PN [cm⁻³] | Reference |
|---|---|---|---|---|---|
| Sierra Nevada Mountains, US | 1315 | May-Nov 2002 | 10-400 | 4300 | (Lunden et al., 2006) |
| Mt. Tai, China | 1534 | July 2010-Feb 2012 | 3-2500 | 11800±6200 | (Shen et al., 2016) |
| Mt. Huang, China | 1840 | April-Aug 2008 | 10-10000 | 2350 | (Zhang et al., 2016) |
| Mukteshwar, India | 2180 | Nov 2005-Nov 2008 | 10-800 | 2730 | (Komppula et al., 2009) |
| Izana Observatory, Spain | 2367 | Nov 2006-Dec 2007 | 3-660 | 480-4600 | (Rodríguez et al., 2009) |
| Mt. Norikura, Japan | 2770 | Sep 2001, July-Sep 2002 | 9-300 | 260-1600 | (Nishita et al., 2008) |
| University of Colorado Mountain Research Station, US | 2900 | July 2006 | 3-800 | 2881-19947 | (Boy et al., 2008) |
| Dome C, Antarctica | 3200 | Spring, 2008-2009 | 10-600 | 17.9-457 | (Järvinen et al., 2013) |
| Storm Peak Laboratory, US | 3210 | Mar 2012 | 10-10000 | 3100 | (Yu and Hallar, 2014) |
| Jungfraujoch, Switzerland | 3580 | 1995-2015 | 10-10000 | 757 | (Bukowiecki et al., 2016) |
| Wangliguan, China | 3816 | Sep 2005-May 2007 | 12-570 | 2030 | (Kivekäs et al., 2009) |
| Mt. Yulong, China | 3410 | May-April 2015 | 3-10000 | 1600±1290 | This Study |


3.1.2 Analysis of PNSD and PVSD

Average of PNSD during the measurement is showed in Fig.2a. In this study, we

sorted the particles by their sizes (Dal Maso et al., 2005). $N_{25-100}$ correlates to primary
emission, and $PN_{100-1000}$ has stronger connection with secondary formation (Wu et al.,
2008). The diameter with highest particle number concentration ($D_{p-max}$) was 107 nm.
Number concentration (dN/dlogDp) was larger than 1000 cm⁻³ between 40-200 nm,
which was the adjacent area of $N_{25-100}$ and $N_{100-1000}$. This indicates that both primary
emission sources and secondary formation process had influences at Mt. Yulong site.



$N_{3\text{-}25}$, $N_{25\text{-}100}$, $N_{100\text{-}1000}$ were 244 cm$^{-3}$, 676 cm$^{-3}$ and 638 cm$^{-3}$, constituting 16 %, 43 %
and 41 % of total concentration, respectively.

Different from PNSD, particle volume (PV) exhibited a bimodal distribution (Fig

2a). The first peak had an extreme value at 340 nm, representing the contribution of
primary emission and aging processes. This mass peak constituted 66 % of total PV,
including PV$_{25\text{-}100}$ (2 %) and PV$_{100\text{-}1000}$ (64 %). 3-25 nm particles had negligible
influence on PV. Another mode in PV size distribution is within range of 1μm-10μm,
with the $D_{p\text{-}max}$ at 2.2 μm. This mode could be attributed to the suspended soil.
Volume of 1-10μm particles constituted 34 % of total PV, similar with Qilian Shan
station (38 %) at Northeast Tibet Plateau (Xu et al., 2015), but higher than that urban
Beijing (25 %) (Wu et al., 2008), due to the much less emission sources and stronger
wind at Mt. Yulong.

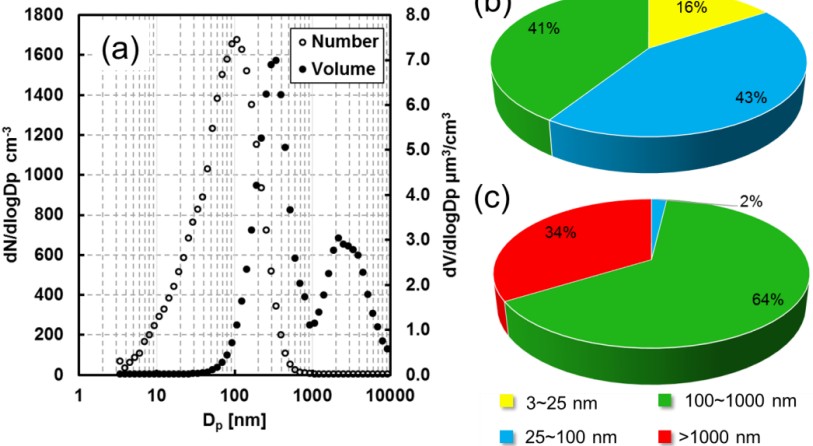


**Figure 2. Particle size distribution in atmosphere at Mt. Yulong. (a) Mean size**
**distribution of particle number (hollow circle) and volume (filled circle)**
**concentration; Contribution of different fractions to total particle (b) number**
**concentration and (c) volume concentration. Different colors represent different**
**size ranges: yellow (3-25 nm), blue (25-100 nm), green (100-1000 nm), red (1-10**
**μm).**




To better characterize the contribution from different process, the mean PNSD
was fitted to three lognormal modes (Fig. 3, Table 3). We define the three fitted modes
as nucleation mode, Aitken mode and accumulation mode, based on their geometric
mean diameters, which were within 3-25 nm, 25-100 nm and 100-1000 nm,
respectively. Nucleation mode can be derived from nucleation process. Nucleation
mode contributed 15 % to total PN, which was half lower than proportion of
nucleation mode particles at Mt. Tai, indicating relatively less impact from nucleation
events. Median diameter of Aitken mode and accumulation mode particles are 52 nm
and 130 nm. These mean diameters are similar with the results at Jungfraujoch
(Bukowiecki et al., 2016) and Beijing (Wu et al., 2008). Accumulation mode particles,
correlating with secondary formation (mode_3), contributed 54 % to total PN, which
is twice higher than the result in urban Beijing (Wu et al., 2008), and similar with that
in pristine atmosphere of Jungfraujoch (Bukowiecki et al., 2016). This result indicates
that aerosols arrived at Mt. Yulong were aged during the transport.

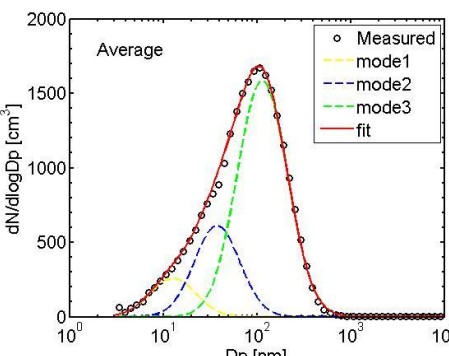


**Figure 3. Lognormal fit (3 modes) of average particle number size distribution**
**during the campaign at Mt. Yulong. Black circles mark the measured PNSD,**
**colored dash lines represent the PNSD of fitting modes, and red full line marks**
**the sum of PNSD of all fitting modes. Mode 1, 2 and 3 were nucleation mode,**
**Aitken mode and accumulation mode, respectively.**





### 3.2 Influence of PBL diurnal variation on PNSD

Figure 4 shows the diurnal variation of $N_{3-25}$, $N_{25-100}$, $N_{100-1000}$ during the sampling period. The particle number concentration in nucleation fraction and Aitken fraction showed a clear diurnal variation. Mean value of $N_{3-25}$ started to increase at 10:00 in the morning, and reached around 500 cm$^{-3}$ at noon due to nucleation events at noon (Fig. 4a). However, the median of $N_{3-25}$ didn't showed similar diurnal variation because of low NPF frequency. On the other hand, both mean and median of $N_{25-100}$ showed a local maximum during 10:00-14:00 (Fig. 4b). NPF events could not cause this variation, since newly formed particles were not able to grow to 25 nm in the morning. So the increased 25-100 nm particles originated from primary sources, e.g. traffic sources and biomass burning. Considering that no anthropogenic emission sources around the site, those primary particles could be transported from other regions. During noon time, as the convection is strongest, $N_{25-100}$ could be raised by the elevated urban PBL during the day, and anthropogenic particle injected during this process (Tröstl et al., 2016b). Adak et al. (2014) also reported that number concentration of PM$_1$ increased during day time, corresponding with the up-slope valley wind. In the afternoon, the convection become weaker, and the larger wind speed (Fig. 4e) had stronger scavenging effect on those primary particles, so $N_{25-100}$ decreased at around 14:00.

The diurnal change of absolute water content also support that Mt. Yulong site was influenced by elevated PBL during midday. The water concentration was calculated based on temperature and relative humidity, and showed an increase from 3.3 g m$^{-1}$ to 4.2 g m$^{-1}$ during 9:00-12:00, and descended back to 3.6 g m$^{-1}$ till 14:00 (Fig. 4f). This systematic water content variation indicates that the site was influenced by the PBL during day time. Shen et al (2016) used the increase of water content together with Aitken mode particles, to separate the PBL conditions at Mt. Tai. The value of CO/NO$_y$ and NO$_y$/NO$_x$ was used in other studies to determine the age of the air masses arriving high altitude sites (Tröstl et al., 2016b;Zellweger et al., 2003;Jaeglé et al., 1998). Because NO$_y$ was not measured in this study, we used





CO/NOₓ to estimate the age of air mass since contact with primary emission. $CO/NO_x$
was 287±146 at Mt. Yulong, lower than Jungfraujoch (Herrmann et al., 2015), Mt.
Cimone (Cristofanelli et al., 2016) and Kansas (Jaeglé et al., 1998), indicating a
stronger anthropogenic influence. The diurnal variation of $CO/NO_x$ showed minimum
during 9:00-14:00 (Fig. 4d), consistent with the local maximum of water content and
$N_{25-100}$. The diurnal variation of $O_3/NO_x$ exhibited a similar trend as $CO/NO_x$, with an
average of 69.6 during 10:00-14:00, and 83.6 during 1:00-6:00 (Fig. S2). Those
evidences indicate that at least during 10:00-14:00, Mt. Yulong site was influenced by
elevated PBL. On the other hand, we consider the data during 1:00-6:00 as the
condition within FT, when $N_{25-100}$ and water content were lowest and $CO/NO_x$ were
highest. However, $N_{100-1000}$ didn't show obvious diurnal variation, indicating the
elevated PBL didn't inject large amount of 100-1000 nm particles.

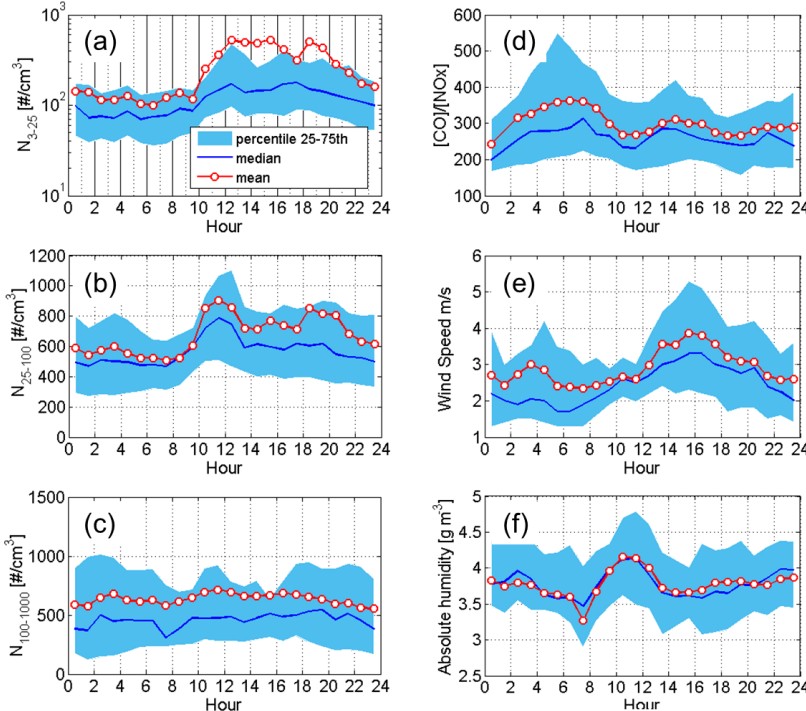


**Figure 4. Diurnal variations of N₃₋₂₅, N₂₅₋₁₀₀, N₁₀₀₋₁₀₀₀, CO/NOₓ, wind speed, and**
**absolute humidity at Mt. Yulong during monitoring campaign. Red lines with**
**circles, blue lines mark the mean and median results, respectively. Light blue**



**area marks the range between 25th, and 75th percentiles of the data.**
3.3 Influences of BB on Mt. Yulong
3.3.1 Identification of BB episodes
Background condition (BG) was picked during 1 to 4 April, when the
concentration of BC was 85 ng m$^{-3}$ on average. During this period, the wind was
relatively stronger, and fire spots were barely found on the westward path of the air
mass (Fig. S3). Based on the background condition, three BB events were identified
by the following criteria: 1) BC was more than the background level (85 ng m$^{-3}$); 2)
higher fraction of $f$60 than during BG (0.4 %); 3) fire spots appeared in the source
regions of the air masses or surrounding areas of the site.
During the first BB event (BB1, 22 to 30 March), dense fire spots were found on
the source region in north Burma. BC concentration (1.2 μg m$^{-3}$), PV and $f$60 signal
showed highest level during BB1. Trajectories of BB2 (5 to 6 April) passed fewer fire
spots in South Asia than BB1, and the BC concentration was lower (0.3 μg m$^{-3}$). The
proportion of $f$60 was highest (1.4 %) during BB3 (11 to 12 April), showing strong
BB influence. However, few fire spots were observed on the path of air mass,
indicating the BB particles could be derived from domestic heating nearby.





### 3.3.2 Influences of BB on PNSD and CCN

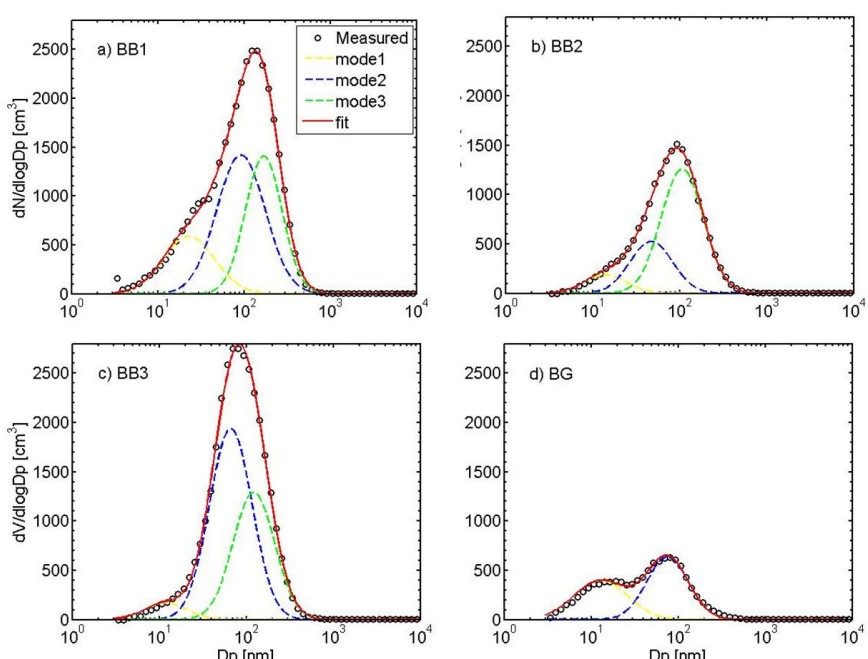

**Figure 5. Lognormal fit (3 modes) of average PNSD during (a) BB1, (b) BB2, (c)**
**BB3, (d) BG at Mt. Yulong site. Black circles mark the measured PNSD, colored**
**dash lines represent the PNSD of fitting modes, and red full line marks the sum**
**of PNSD of all fitting modes. Mode 1, 2, 3 were nucleation mode, Aitken mode**
**and accumulation mode, respectively.**
The average PNSDs during BB events and BG condition were plotted in Figure 5,
and fitted by 3-mode lognormal distributions. The fitting results are showed in Table 3.
For the BG condition, only two modes were obtained (Fig. 5d), including nucleation
mode (mode_1, $D_{mean}$ = 15 nm) originated from nucleation events and Aitken mode
(mode_2, $D_{mean}$ = 79 nm). Their total number concentration was 669 cm$^{-3}$, similar
with the level at Swiss Jungfraujoch (Herrmann et al., 2015), exhibiting an Eurasia
background character.
The averaged PNSD during those BB episodes showed discrepancies, indicating
the variant influences of transported and local BB on particles in atmosphere of Mt.



Yulong. During the BB2, Aitken mode number concentration was similar with that
during BG condition. But an accumulation mode (mode_3, $D_{mean}$ = 106 nm) with
higher PN (775 cm$^{-3}$) appeared (Fig. 5b). This mode with larger size could be the aged
particles transported from BB source regions in South Asia. Different from BB2,
Aitken mode particles were increased by a factor of 3 and became dominant in PNSD
during BB3 the local event (Fig. 5c). The number concentration of this mode was
1309 cm$^{-3}$, 2 times more than accumulation mode particles. The reason could be that
the particles during BB3 were freshly emitted from sources nearby the monitoring site.
The study of Zheng et al (2017) also showed that during this period, the OOA fraction
in organic aerosols was relatively lower, while BBOA fraction was higher, indicating
impacts from more local BB sources. The geometric mean diameters of accumulation
mode were 169, 106, 130 nm during BB1, BB2, and BB3, smaller than that of aged
biomass burning particles at Mt. Bachelor, USA (Laing et al., 2016), indicating the
particles at Mt. Yulong were more fresh. Nucleation mode had lower PN during BB2
and BB3, since the higher PN of larger particles played as strong coagulation sink of
nucleation mode particles. Aitken mode and accumulation mode were comparable
during BB1 (Fig. 5a), indicating the fresh aerosols from sources surrounding the site
had comparable influence as the transported aged BB aerosols.

In a word, the BG condition at Mt. Yulong could represent the background level

of particles of TP or even Eurasia. The local and long-range transported BB emissions
would increase the level of Aitken mode and accumulation mode particles,
respectively.










**Table 3. Fitted parameters of lognormal modes for different period. μ, σ and N represent the mean diameter, standard deviation, and total number concentration of each mode, respectively. "Total" represents the mean result of all data achieved from the campaign.**

| Period | μ [nm] | | | σ [nm] | | | N [cm$^{-3}$] | | |
|--------|--------|--------|--------|--------|--------|--------|--------|--------|--------|
| | mode_1 | mode_2 | mode_3 | mode_1 | mode_2 | mode_3 | mode_1 | mode_2 | mode_3 |
| Total | 16 | 52 | 130 | 1.75 | 1.75 | 1.77 | 221 | 488 | 861 |
| BB1 | 23 | 92 | 169 | 1.94 | 1.91 | 1.63 | 428 | 1014 | 744 |
| BB2 | 16 | 47 | 106 | 1.75 | 1.75 | 1.75 | 117 | 302 | 775 |
| BB3 | 15 | 70 | 130 | 1.75 | 1.75 | 1.72 | 106 | 1309 | 628 |
| BG | 15 | 79 | - | 2.03 | 1.73 | - | 301 | 368 | - |

Concentration of CCN was calculated following method described in section 2.3.3. κ value during the sampling period was 0.12±0.01, only 1/3 from urban Beijing (Wu et al., 2016) and a rural site at Thuringia, Germany (Wu et al., 2013), but consistent with the results in Alberta, Canada (0.11±0.04) during BB events (Lathem et al., 2013). Pierce et al (2012) reported that κ was around 0.1 for >100 nm particles in a forest mountain valley during biogenic secondary organic aerosols formation and growth events. Similarly, organic volume fraction was 0.73 in particle at Mt. Yulong, explaining the low value of κ. As a result, the $D_c$ at SS of 0.6 % and 1.2 % was 72.0±2.2 and 45.4±1.4 nm, respectively. There could be uncertainties for value of κ and $D_c$, since here we used a manually set hygroscopicity of organics, which may varied with oxidation level or other factors (Wu et al., 2016). Considering the variation range of $D_c$ was small, the CCN concentration was mainly controlled by size distribution of particle number.

BB events raised the CCN level in atmosphere by influencing the PNSD. Increase of PN was observed during BB events, i.e. 2207±1388 cm$^{-3}$, 1214±638 cm$^{-3}$, 2062±1112 cm$^{-3}$ during BB1, BB2, BB3, respectively. As a consequence, the increased particles played as CCN in atmosphere of Mt. Yulong, forming a readily increase of CCN concentration during BB events. Figure 6 showed the mean number concentration of CCN in periods under PBL influence (10:00-14:00, as discussed in 3.2) and FT condition (1:00-6:00) during BG and BB events. Mean number



concentration of CCN under supersaturation of 0.6 % was 936±754 cm⁻³ and 807±705
cm⁻³ for PBL and FT periods at Mt. Yulong, comparative with boreal forest station in
Finland (Cerully et al., 2011). The concentration of CCN in PBL condition during
BB1, BB2, BB3 was 5, 2, 2 times as that during BG. Promotions of CCN during BB1,
BB2, BB3 were more remarkable for FT, i.e. 9, 3, 8 times as BG (Fig. 6). This result
indicates that the BB particles from South Asia could have strong influence on the
climate parameter. For the data under supersaturation of 1.2 %, the ratios between
CCN concentration of BBs and that of BG were less, i.e. 2-4 times for periods
influenced by PBL, and 2-7 times for FT conditions. This is because critical diameters
under supersaturation 0.6 % and 1.2 % were around 72 nm and 45 nm, respectively.
And PN within 45-72 nm was relatively stable compared to the larger particles,
because of the daily input of anthropogenic primary aerosols from urban air masses.

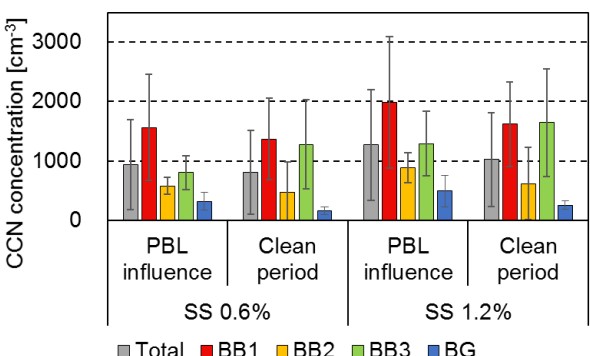


**Figure 6. Mean number concentration of CCN under supersaturation of 0.6 %**
**and 1.2 % during whole monitoring period (labelled as "total"), BB1, BB2, BB3**
**events (marked by shadow in Fig. 1). PBL (10:00-14:00) and FT (1:00-6:00)**
**conditions were separated.**

3.4 New particle formation events
3.4.1 NPF events at Mt. Yulong under anthropogenic influences
Following the method Tröstl et al (2016b) and Yli-Juuti et al (2009) used, we



define the three NPF events on 29 March, 4 and 13 April as follows:
a) Type A event on 29 March: appearance of newly formed particles (3 nm, the first

bin of nano-SMPS) and continuous growth of those particles, reaching the upper

limit of nucleation mode (25 nm). This NPF event was within the period of BB1,

the air mass arriving at Mt. Yulong was from North part of Burma with slow

movement before 28 March, transporting abundant pollutants to this area. During

28 March 12:00 to 29 March 6:00, the air mass was from west upper troposphere

(Fig S4), cleaning out the pre-existing particles built up by BB events. CS at Mt.

Yulong decreased from 0.006 $s^{-1}$ to 0.002 $s^{-1}$ on morning of 29 March. On the

other hand, the concentration of $SO_2$ was stable (around 0.04 ppb) before

occurrence of nucleation. The calculated $H_2SO_4$ increased before nucleation,

reaching $5 \times 10^6$ $cm^{-3}$ (Fig. S4). The nucleation rate was 1.43 $cm^{-3}s^{-1}$, increasing

nucleation mode particles to around $10^4$ $cm^{-3}$. Benefited by the increase of

α-pinene (from 0.02 ppb to 0.10 ppb), β-pinene (from 0.03 ppb to 0.20 ppb) and

$SO_2$ in day of 29 March, the formation of secondary aerosol continued, and the

newly formed particles grew over 30 nm before night. This process could be from

gaseous oxidation and condensation. Particle growth stopped during night of 29

March, when the gaseous reaction was inhibited because of absence of sunlight.

After sunrise on 30 March, the particle continued growing and reached 40 nm.

Concentration of toluene was lower than 0.1 ppb, indicating small contribution

from anthropogenic VOCs. The GR was 3.48 nm $h^{-1}$ within size range of 3-25 nm.

In a word, under the influence of transported pollutants, nucleation was triggered

by the upper clean air mass, which reduced level of pre-existing particles. While

growth of particles was favored by the photochemical reaction and condensation

process.


b) Type B event on 4 April: newly formed mode occurred with growing trend, but

growth stopped at early stage (<15 nm), and there were temporal low values of

$N_{3-25}$ during the event. The event on 4 April was under BG condition, during

which concentration of $SO_2$ was lower (around 0.02 ppb). Started from 2:00 on 4



April, the air mass arrived at Mt. Yulong was transported by upslope flow from
west lower troposphere. While during 9:00-12:00, the air mass arrived at Mt.
Yulong passed Northeast India, where fire spots could be observed on the MODIS
map during 2 April. Thus, the gaseous pollutants and particles from anthropogenic
sources nearby or from BB sources in Northeast India was transported to this site
on morning of 4 April. Concentration of $SO_2$ and $[H_2SO_4]$ increased to 0.07 ppb
and $6 \times 10^6$ $cm^{-3}$ at 11:00, respectively, corresponding to occurrence nucleation (Fig.
S5). $SO_2$ shared similar time series with black carbon, indicating a combustion
source. FR and GR were 0.93 $cm^{-3}$ $s^{-1}$ and 3.2 nm $h^{-1}$, respectively. $N_{3-25}$ fluctuated
during NPF events, showing low values when there were temporary changes in
cloud condition (influencing radiation) and wind direction. Concentrations of
β-pinene and Toluene were stable and lower than 0.10 ppb and 0.65 ppb
respectively throughout the NPF event, which could be the reason of smaller
growth rate. At around 14:00, the source region of air mass varied to west upper
troposphere, and the stronger wind cleaned out both the nucleated mode particles
and gaseous precursors, terminating the NPF event. In summary, type B event
under background condition was triggered by the injection of gaseous pollutants
from elevated PBL and short term transport of BB pollutants.

c)  Off-site NPF event on 13 April: A narrow Aitken mode band (25-50 nm) was
observed from 13 to 14 April. The primary particles should have wider range and
larger size. These particles were mostly likely nucleated off-site, and transported
to Mt. Yulong site by uplifting air mass. On the afternoon of 13 April, the air mass
arrived at Mt. Yulong passed the local ground layer (yellow trajectory in Fig S6).
$SO_2$ increased from 0.06 ppb to 0.13 ppb, and toluene reached highest level at
0.098 ppb (Fig. S6), indicating an anthropogenic impact. As a result, particles
formed from the ground level were transported to the site and a burst of $N_{3-25}$
occurred at around 18:00, with FR at 1.64 $cm^{-3}$ $s^{-1}$. β-pinene also showed higher
value at dawn of 13 April. Those nanoparticles showed a growth trend, with GR at
2.99 nm $h^{-1}$. To summarize, occurrence of nucleation mode particles were off-site



nucleated in PBL and transported to this site.

3.4.2 Limiting factors of NPF events
Frequency of NPF was 14 % during our measurement. This NPF frequency is
clearly less than polluted atmosphere of North China Plain (40-65 %) in March and
April (Wang et al., 2013;Shen et al., 2011), the top of Mt. Huang (38 %) during April
(Zhang et al., 2016), and a remote rural site in the Sierra Nevada Mountains (47 %) in
spring (Creamean et al., 2011). A common knowledge is that CS is the limiting factor
that controls the NPF (Cai et al., 2017). Thus, pre-existing particle levels on event
days should be less than non-event days, at high altitude mountain sites (Shen et al.,
2016;Guo et al., 2012) as well as urban sites (Wang et al., 2011;Wang et al., 2017).
The low NPF frequency was unexpected in clean atmosphere of Mt. Yulong, since the
mean CS at Mt. Yulong was 0.0038 s$^{-1}$. On the other hand, similar low frequency of
NPF events were also observed in pristine atmospheres, e.g. 24 % at Antarctic site
Neumayer (Weller et al., 2015), 12-17 % at Dome C, Antarctica (Järvinen et al.,

2013).

During the first five days of the campaign (22 to 27 March), the nucleation
events could be prevented by large amount of pre-existing particles acting as big
condensation sink. The CS was more than 0.005 s$^{-1}$, similar with polluted Beijing on
days with NPF events (Wu et al., 2007). However, on rest of days when CS was even
lower than 0.002 s$^{-1}$, the NPF events were still scarce. Considering that the content of
condensable vapor participated in nucleation is determined by the competition
between formation from precursor oxidation and condensation on surface of
pre-existing particles (Zhang et al., 2012), the lower NPF frequency at pristine sites
could be resulted from lack of precursor, e.g. VOCs and $SO_2$ from fossil fuel and
biomass burning sources.
To further evaluate the effect of different parameters on NPF, daily variations of
$SO_2$, CS, $J(O^1D)$, Benzene and β-pinene during 28 March to 14 April were calculated
and plotted in Figure 7. The results during 10:00-14:00 were picked up as the





occurrence time of nucleation, and compared between NPF days and non-event days.
As shown in Fig. 7, NPF days and non-NPF days shared same level of $J(O^1D)$, and
15 % difference in CS when nucleation happened, indicating small influence of solar
radiation and pre-existing particles on NPF.
The concentration of $SO_2$ showed increase on NPF days, 60 % higher than
non-event days, indicating the anthropogenic $SO_2$ as the controlling factor of NPF at
Mt. Yulong. Studies at Jungfraujoch (Bianchi et al., 2016;Tröstl et al., 2016b), Izaña
(Garcá et al., 2014) and Mukteshwar (Neitola et al., 2011) also reported that the
nucleation events in upper troposphere corresponded to increase of anthropogenic gas
pollutants by elevated PBL. At Daban Mountain on the North slope of TP, the $PM_{2.5}$
level was similar with Mt. Yulong, but NPF could be observed nearly every day. It
may be caused by that the $SO_2$ was around 2 ppb on average, two order of magnitude
higher than Mt. Yulong.
Organics may also be a driven factor on NPF. The concentration of β-pinene
showed higher value (40 %) in the afternoon on NPF days, while there was little
difference (9 %) between NPF days and non-event days on anthropogenic benzene.
Recent studies considered that apart from sulfuric acid, the highly oxidized
multifunctional organics from biogenic VOCs could take part in nucleation as well as
growth (Huang et al., 2016;Tröstl et al., 2016a), in free troposphere, the pure organic
nucleation without sulfuric acid may even be dominant (Bianchi et al., 2016;Gordon
et al., 2016). So the increase of biogenic VOCs could benefit the nucleation and
growth of nucleation mode particles at Mt. Yulong.




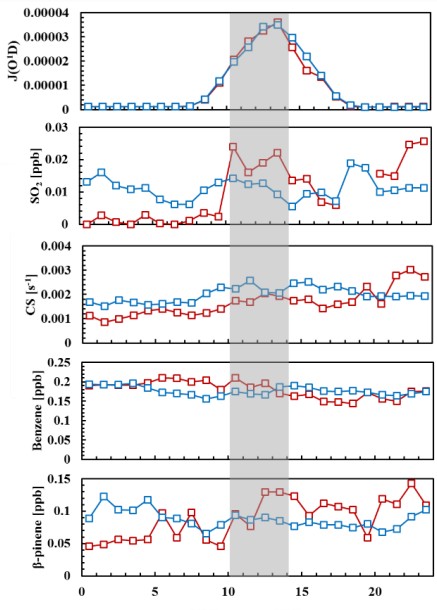


**Figure 7. Diurnal variation of $J(O^1D)$, $SO_2$, CS, Benzene, β-pinene during days
with NPF events (labelled as "NPF", red lines with marks), and without NPF
events (labelled as "non-NPF", blue lines with marks). Shadow marks the time
period during which nucleation occurred.**


3.4.3 Parameters of NPF events at Mt. Yulong
Formation rate, growth rate and condensation sink of NPF events at Mt. Yulong
were summarized in Table 4. Compared with other high mountain measurements, this
study reported a higher FR, e.g. 3 times as that at Storm peak laboratory (Hallar et al.,
2011). But the GR at Mt. Yulong was within average level, indicating different
precursors participating in nucleation and growth process. Even though $SO_2$ was well
correlated with nucleation events, the calculated growth rate by condensation of
$H_2SO_4$ can only explain 5 % of the measured GR. This result indicated participation
of some other precursors in particle growth, e.g. organics.




**Table 4. Comparisons of NPF parameters (FR, GR, CS) with the other studies.**

| Site | Region | Altitude [m] | Size [nm] | FR [cm⁻³s⁻¹] | GR [nm h⁻¹] | CS [s⁻¹] | Reference |
|------|--------|--------------|-----------|--------------|-------------|----------|-----------|
| Mt. Yulong | Asia | 3410 | 3-25 | 1.33 | 3.22 | 0.002 | This research |
| Mukteshwar | Asia | 2180 | 15-20 | 0.44 | 2.47 | 0.015 | (Neitola et al., 2011) |
| Storm peak laboratory | North America | 3210 | 9-334 | 0.39 | 7.5 | 0.001 | (Hallar et al., 2011) |
| Mt. Tai | Asia | 1500 | 3-25 | 4 | 6.1 | 0.02 | (Shen et al., 2016) |
| Izaña | Atlantic Ocean | 2400 | 10-25 | 0.46 | 0.43 | 0.002 | (García et al., 2014) |
| Jungfraujoch | Europe | 3580 | 3.2-15 | 1.8 | 4.0 | - | (Tröstl et al., 2016b) |
| Dome C | Antarctica | 3200 | 10-25 | 0.023 | 2.5 | 0.0002 | (Järvinen et al., 2013) |


**4.** Conclusion

PNSD, meteorological parameters, trace gases and particle chemical composition

were measured at Mt. Yulong site (3410 m a.s.l.) in Southeast corner of Tibet Plateau,
during pre-monsoon season (22 March to 15 April) of 2015. PNSD in background
atmosphere of Tibet Plateau was characterized. As a background site in Southwest
China, the atmosphere of Mt. Yulong exhibited a feature of low particle level and
strong oxidation.

PBL convection is an influencing factor of PNSD, which caused readable diurnal

variation of $N_{ait}$. Diurnal variation of CO/$NO_x$ and absolute humidity showed that the
monitoring site was influenced by PBL during 10:00-14:00, and showed typical FT
condition during 1:00-6:00.

Three different types of BB event periods were identified by content of BC, $f60$,

air mass backward trajectory and fire spot map. Accumulation mode was dominant in
transported BB particles from Myanmar, but less aged compared with other Tibet
Plateau sites under influence of BB. Under local biomass burning episode, Aitken
mode was dominant in PNSD. The biomass burning from South Asia had strong
influence on climate parameters, especially for FT. Concentrations of CCN in FT at
Mt. Yulong during BB events were 3-9 times as that during BG period. Due to high
fraction of organic compounds, the CCN activity of particles in atmosphere of Mt.
Yulong was lower than other high altitude sites and ground level sites.





Unexpected low NPF frequency was found in clean atmosphere at Mt. Yulong,
due to low concentration of anthropogenic precursor, i.e. $SO_2$. Occurrence of NPF
events were favored by elevated surface emission of $SO_2$ and transported BB
pollutants from South Asia. Off-site NPF event was also observed, during which
nanoparticles were formed in PBL and transported to the site. Condensation of
sulfuric acid can only explain 5 % of GR in on-site NPF events, indicating other
precursors participating in particle growth. NPF can hardly contribute to CCN, since
the newly formed particles cannot reach the critical diameter.
Our study provided important data in vertical profile of particles at Tibet Plateau.
Influences of BB activities in South Asia and local area on PNSD and CCN in
atmosphere of Tibet Plateau were highlighted. Different types of NPF in upper
troposphere in Southwest China were characterized, and role of $SO_2$ were analyzed.
Results of our study could be used in regional and global climate model, and help
building up the knowledge of NPF in upper part of troposphere.

**Acknowledgement**
This study was supported by National Natural Science Foundation of China
(91544214, 41421064, 51636003, 21677002), National Key Research and
Development Program of China (2016YFC0202000: Task 3), the China Ministry of
Environmental Protection Special Funds for Scientific Research on Public Welfare
(201309016). We also thank the colleagues in the China National Environmental
Monitoring Center and Lijiang Monitoring Station for the support to the field
campaign.

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
