# Peer review of "Particle number size distribution and new particle formation under influence of"

_Atmospheric Chemistry and Physics, 2018_

## Referee Comment (RC1) · Anonymous Referee #1 · 23 Apr 2018

The manuscript by Shang et al. reports the particle number size distributions (PNSD) and new particle formation at a high altitude background site in China. The influences of biomass burning on PNSD and cloud condensation nuclei are investigated, and three new particle formation events were characterized. The sampling site (3410 m) is unique, and the data is also potentially important for validation of climate models. This manuscript is generally well written, and fits within the scope of ACP. I recommend it for publication after addressing the following comments.

Comments:

1. Identification of BB2 needs more evidence. As indicated in Figure 1, $f_{60}$ is very close to that during the background period. Also, the back trajectory analysis in supplementary did not show a strong influence of biomass burning on the sampling site.
2. While discussing the average particle number size distributions, could the authors show the average PNSD during NPF events and non-NPF events. As shown in Figure 1, the three NPF events show very high concentrations of particles between 3 – 25 nm, which are rarely seen during non-NPF days.
3. Please describe the instruments for measuring gaseous species, e.g., $SO_2$, CO, NO, $NO_x$ etc.  Because the concentrations of several gaseous species are very low (e.g., < 0.3 ppb for $SO_2$), the measurement uncertainties could be large.
4.  Some analysis in this work can be more robust by incorporating the HR-ToF-AMS data which is published in Zheng et al. (2017) from the same group.
5. Suggest adding  "number" in the title, which is "Particle number size distribution".
6. Line 635, this study did not provide vertical profile of particles.
7. The results can also be compared with another mountain site (3295 m, ASL)  in Tibet Plateau (Du et al., 2015).

Reference:

Du, W., Sun, Y. L., Xu, Y. S., Jiang, Q., Wang, Q. Q., Yang, W., Wang, F., Bai, Z. P., Zhao, X. D., and Yang, Y. C.: Chemical characterization of submicron aerosol and particle growth events at a national background site (3295 m a.s.l.) on the Tibetan Plateau, Atmos. Chem. Phys., 15, 10811-10824, 10.5194/acp-15-10811-2015, 2015.

---

## Referee Comment (RC2) · Anonymous Referee #2 · 27 Apr 2018

This manuscript reports the results of particle aerosol size distribution and new particle formation at a high altitude background site of the southeast Tibet-Plateau. This study also analyzes the influences of biomass burning on particle size distribution and potential cloud condensation nuclei. Based on the co-located measurements such as VOCs, SO2, NOx, CO and so on, the influence factors on NPF were also discussed. This manuscript is suitable to be published in ACP after revision. There are some specific comments for authors:

1. Instruments: there are two set of scanning mobility particle sizer (SMPS) and an

aerodynamic particle sizer (APS) used for PNSD. When combined these data, how to deal with the overlap size range, especially for two SMPS? 2. About backward trajectory analysis, why use different models for that, what's the difference? 3. Figure 4a, there are something wrong, the mean value is not in the range of 25-75% percentile. 4. About New particle formation events: there are only three so-called NPFs during this observation period. One of them is defined by authors "Off-site NPF" during which nanoparticles were formed in PBL and transported to the site and a burst of N3-25 occurred at around 18:00, with FR at 1.64 cm-3s-1. This is contradicted. All three NPFs have different characteristics. The parameters such as FR, GR, CS may not representative for this region statistically. 5. In conclusion: Such points are not discussed in the manuscript, but in the conclusion, such as "the atmosphere of Mt. Yulong exhibited a feature of strong oxidation", "Our study provided important data in vertical profile of particles at Tibet Plateau" 6. The elevation of sampling site: 3140m or 3410m? 7. Line 106: Silicon diffusion tube?

---

## Referee Comment (RC3) · Anonymous Referee #3 · 2 May 2018

This is a well prepared paper on the particle size distribution and NPF at a high elevated background site of China. My only concern is that the author indicated that N25-100 correlates to primary emission, e.g. traffic sources and biomass burning and PN100-1000 has stronger connection with secondary formation. Also, the author attributed the extreme value at 340 nm to the contribution of primary emission and aging processes. The paper Wu et al., 2008 was cited. However, Wu et al. (2008) indicated that "Laboratory studies showed that mean diameters for the number size distributions of particles emitted by gasoline engines ranged from 40 nm to 80 nm" and "Their results showed that geometric mean diameters of particles emitted by all kinds of biofuels combustion were in the range from 110 nm to 200 nm." So the author should cited the emission test results for different kinds of fuels burning directly (diesel and gasoline traffic, biomass burning, coal, etc.) and re-write corresponding sentences.

---

## Author Comment (AC1) · 11 Jul 2018

We thank referee's efforts in reviewing our manuscript. Following are the replies to the comments:

Response to referee comments #1 :

Comment 1. Identification of BB2 needs more evidence. As indicated in Figure 1, f60 is very close to that during the background period. Also, the back trajectory analysis in supplementary did not show a strong influence of biomass burning on the sampling

site. Reply According to the analysis of this study and Zheng's(2017), the sampling site was influenced by oxidative biomass burning particles during BB2. The decisive evidence in identifying BB2 is the elevated concentration of BC, from 115 ng to around 500 ng at night of April 5th. On the fire map, several, if not a lot, fire spots appeared around the start and middle of the back trajectory of BB2. The f60 ratio showed several peaks (> 1%) during BB2 period, and the average was 0.61% slightly higher than that during background period (0.40%).

Comment 2. While discussing the average particle number size distributions, could the authors show the average PNSD during NPF events and non-NPF events. As shown in Figure 1, the three NPF events show very high concentrations of particles between 3 – 25 nm, which are rarely seen during non-NPF days. Changes in manuscript. One figure was added in the supplementary, showing the averaged PNSD during NPF events and the non-NPF event periods. To eliminate the influences from biomass burning, another average PNSD during the periods without either NPF events or biomass burning influences was included. The differences between the those PNSDs were discussed added in section 3.4.1 and section 3.4.2.

Comment 3. Please describe the instruments for measuring gaseous species, e.g., SO2, CO, NO, NOx etc. Because the concentrations of several gaseous species are very low (e.g., < 0.3 ppb for SO2), the measurement uncertainties could be large. Changes in manuscript. Description and uncertainty of trace gas detection was added in section 2.2.

Comment 4. Some analysis in this work can be more robust by incorporating the HR-ToF-AMS data which is published in Zheng et al. (2017) from the same group. Changes in manuscript. More evidences in identifying the BB events were taken from that AMS data set in section 3.3.1.

Comment 5. Suggest adding "number" in the title, which is "Particle number size distribution". Changes in draft Title was changed following the comment.

Comment 6. Line 635, this study did not provide vertical profile of particles. Changes in manuscript. This sentence was changed as "Our study provided important dataset in vertical profile of particles physical properties at Tibet Plateau."

Comment 7. The results can also be compared with another mountain site (3295 m, ASL) in Tibet Plateau (Du et al., 2015). Changes in manuscript. Particle mass concentration, NPF frequency and particle number concentration are cited from this paper in the revised version.

Response to referee comments #2:

Comment 1. Instruments: there are two set of scanning mobility particle sizer (SMPS) and an aerodynamic particle sizer (APS) used for PNSD. When combined these data, how to deal with the overlap size range, especially for two SMPS? Reply. Firstly, the data from different instruments for overlap size bins didn't show big difference (Fig. R1). About how we deal with the overlap data: 1) Between two SMPS: We trust the number distribution between (3~60 nm) from the nano SMPS (DMA 3085 and CPC 3776). But the measured sample flow for CPC 3776 is very low (0.05 L/min), and the absolute number concentration has a big uncertainty due to flow fluctuation. And we trust the absolute results from normal SMPS (DMA 3081 and CPC 3022, 0.3 L/min) In order to eliminate the system error from flow of CPC 3776, we calculate the ratio between particle number concentration at the adjacent area (around 60 nm) measured by two set of SMPS for every sample data. Then we use these ratios to correct the number concentration for all size bins for nano SMPS data. 2) Between SMPS and APS: APS data and normal SMPS data had similar value at the overlap diameter (690 nm), that over 76% data is within one order of magnitude. Considering the low number concentration in this size range and uncertainties from OPC detection and diameter transformation from aerodynamic diameter to Stokes dimeter (density assumed as a constant) , this data set is acceptable. So we directly use the SMPS data in diameter range less than 700 nm and APS data for particles larger than 700 nm in Stokes diameter.

Fig R1. PNSD measured by nano SMPS (3-60 nm), normal SMPS (15-700nm) and APS (0.4-10 $\mu$m) before combined. Data were averaged for whole sampling period.

Comment 2. About backward trajectory analysis, why use different models for that, what's the difference? Reply. The WRF model was run in Zheng et al's study (2017) for identification of biomass burning identification. HYSPLIT model was run by Dongjie Shang for analysis of new particle formation. It's just different the choices of the researchers.

Comment 3. Figure 4a, there are something wrong, the mean value is not in the range of 25-75% percentile. Reply. We carefully checked the data , and the results are not wrong. Because mean value is not like media value, sometimes it could be outside the 25%-75% percentile, when the 0-25% or 75-100% values have big difference with other data. For this case, N3-25 during the 3 noon of NPF days are 1-2 order of magnitude higher than other 20 days. Those data are within 75%-100%, but have a very high impact on mean value of N3-25. So we also showed median values in order to give the another representative diurnal variation.

Comment 4. About New particle formation events: there are only three so-called NPFs during this observation period. One of them is defined by authors "Off-site NPF" during which nanoparticles were formed in PBL and transported to the site and a burst of N3-25 occurred at around 18:00, with FR at 1.64 cm-3s-1. This is contradicted. All three NPFs have different characteristics. The parameters such as FR, GR, CS may not representative for this region statistically. Reply. We agree that only 3 individual events could not give representative NPF parameters for this region. Here in this study we want to give the case-based analysis for NPF events we observed, and try to find the key influencing factors. More multi-parameter and long-term studies are required to fully characterize the NPF events in Tibet Plateau and analyze the mechanisms behind them. We define the third event as "off-site NPF" event, considering that the growing particle mode did not start from the lower limit of the detection (3 nm). Since 3-25 nm particles were not formed on site, we no longer talked about FR for this case.

Comment 5. In conclusion: Such points are not discussed in the manuscript, but in the conclusion, such as "the atmosphere of Mt. Yulong exhibited a feature of strong oxidation", "Our study provided important data in vertical profile of particles at Tibet Plateau" Reply and changes in manuscript. We apologize for the confusion. The "strong oxidation" is now changed to "stronger oxidation capacity than low attitude atmosphere", which was described in section 3.1.1. "vertical profile" is not changed as "dataset in vertical profile of particle physical properties".

Comment 6. The elevation of sampling site: 3140m or 3410m? Reply. 3410m, that was a typing error and was corrected.

Comment 7. Line 106: Silicon diffusion tube? Reply. Thanks for stating that, it's now corrected in the manuscript.

Response to referee comments #3 :

Comment. The paper Wu et al., 2008 was cited. However, Wu et al. (2008) indicated that "Laboratory studies showed that mean diameters for the number size distributions of particles emitted by gasoline engines ranged from 40 nm to 80 nm" and "Their results showed that geometric mean diameters of particles emitted by all kinds of bio-fuels combustion were in the range from 110 nm to 200 nm." So the author should cited the emission test results for different kinds of fuels burning directly (diesel and gasoline traffic, biomass burning, coal, etc.) and re-write corresponding sentences. Reply. Thanks for stating that. We rewrite those sentences, and cited results from emission test experiments as well as the PNSD source apportionment studies.

Zheng, J., Hu, M., Du, Z., Shang, D., Gong, Z., Qin, Y., Fang, J., Gu, F., Li, M., Peng, J., Li, J., Zhang, Y., Huang, X., He, L., Wu, Y., and Guo, S.: Influence of biomass burning from South Asia at a high-altitude mountain receptor site in China, Atmos. Chem. Phys., 17, 6853-6864, 10.5194/acp-17-6853-2017, 2017.

[Figure]

**Fig. 1.**

---

## Author Response (AR1)

We thank referee's efforts in reviewing our manuscript. Following are the replies to the comments and changes in the manuscript:

Response to referee comments #1 :

**Comment 1.** Identification of BB2 needs more evidence. As indicated in Figure 1, $f_{60}$ is very close to that during the background period. Also, the back trajectory analysis in supplementary did not show a strong influence of biomass burning on the sampling site.

**Reply** According to the analysis of this study and Zheng's(2017), the sampling site was influenced by oxidative biomass burning particles during BB2. The decisive evidence in identifying BB2 is the elevated concentration of BC, from 115 ng to around 500 ng at night of April 5th. On the fire map, several, if not a lot, fire spots appeared around the start and middle of the back trajectory of BB2. The f60 ratio showed several peaks ($> 1\%$) during BB2 period, and the average was 0.61% slightly higher than that during background period (0.40%).

**Changes in manuscript.** More evidence was added in Section 3.3.1 for identification of BB2. (Line 398-402 in marked up version, Line 355-359 in revised version).

**Comment 2.** While discussing the average particle number size distributions, could the authors show the average PNSD during NPF events and non-NPF events. As shown in Figure 1, the three NPF events show very high concentrations of particles between 3 – 25 nm, which are rarely seen during non-NPF days.

**Changes in manuscript.** One figure was added in the supplementary, showing the averaged PNSD during NPF events and the non-NPF event periods. To eliminate the influences from biomass burning, another average PNSD during the periods without either NPF events or biomass burning influences was included. The differences between the those PNSDs were discussed added in section 3.4.1 and section 3.4.2 (Line 553-559, 591-593 in marked up version, Line 491-497, 527-529 in revised version).

**Comment 3.** Please describe the instruments for measuring gaseous species, e.g., $SO_2$, CO, NO, $NO_x$ etc. Because the concentrations of several gaseous species are very low (e.g., $< 0.3$ ppb for $SO_2$), the measurement uncertainties could be large.

**Changes in manuscript.** Description and uncertainty of trace gas detection was added in section 2.2 (Line 128-138 in marked up version, Line 127-136 in revised version).

**Comment 4.** Some analysis in this work can be more robust by incorporating the HR-ToF-AMS data which is published in Zheng et al. (2017) from the same group.

**Changes in manuscript.** More evidences in identifying the BB events were taken from that AMS data set in section 3.3.1 (Line 398-402 in marked up version, Line 355-359 in revised version).

**Comment 5.** Suggest adding "number" in the title, which is "Particle number size distribution".

**Changes in manuscript.** Title was changed following the comment.

**Comment 6.** Line 635, this study did not provide vertical profile of particles.

**Changes in manuscript.** This sentence was changed as "Our study provided important dataset in vertical profile of particles physical properties at Tibet Plateau." (Line 664-665 in marked up version, Line 594 in revised version)

**Comment 7.** The results can also be compared with another mountain site (3295 m, ASL) in Tibet Plateau (Du et al., 2015).

**Changes in manuscript.** Particle mass concentration, NPF frequency and particle number concentration are cited from this paper in the revised version (Table 2, Line 230, 564-565 and 613-615 in marked up version, Line 224, 503 and 549 in revised version).

Response to referee comments #2:

**Comment 1.** Instruments: there are two set of scanning mobility particle sizer (SMPS) and an aerodynamic particle sizer (APS) used for PNSD. When combined these data, how to deal with the overlap size range, especially for two SMPS?

**Reply.**

Firstly, the data from different instruments for overlap size bins didn't show big difference (Fig. R1). About how we deal with the overlap data:

1) Between two SMPS: We trust the number distribution between (3~60 nm) from the nano SMPS (DMA 3085 and CPC 3776). But the measured sample flow for CPC 3776 is very low (0.05 L/min), and the absolute number concentration has a bigger uncertainty than CPC3022 due to flow fluctuation. So we trust the absolute results from normal SMPS (DMA 3081 and CPC 3022, 0.3 L/min) at the adjacent size range. In order to eliminate the system error from flow of CPC 3776, we calculate the ratios of the particle number concentration measured by two set of SMPS within the last size bin of nano-SMPS (mean diameter at 60.4 nm). Then we use these ratios to correct the number concentration for all size bins measured by nano-SMPS.

2) Between SMPS and APS: APS data and normal SMPS data had similar value at the overlap diameter (690 nm), that over 76% data is within one order of magnitude. Considering the low number concentration in this size range and uncertainties from OPC detection and diameter transformation (from aerodynamic diameter to Stokes dimeter, assuming density as a constant), this data set is acceptable. So we directly use the SMPS data for <700 nm part and APS data for particles larger than 700 nm in Stokes diameter.

[Figure]

Fig R1. PNSD measured by nano SMPS (3-60 nm), normal SMPS (15-700nm) and APS (0.4-10 μm) before combined. Data were averaged for whole sampling period.

**Comment 2.** About backward trajectory analysis, why use different models for that, what's the difference?

**Reply.** The WRF model was run in Zheng et al's study (2017) for identification of biomass burning identification. HYSPLIT model was run by Dongjie Shang for analysis of new particle formation. It's just different the choices of the researchers.

**Comment 3.** Figure 4a, there are something wrong, the mean value is not in the range of 25-75% percentile.

**Reply.** We carefully checked the data , and the results are not wrong. Because mean value is not like media value, sometimes it could be outside the 25%-75% percentile, when the 0-25% or 75-100% values have big difference with other data. For this case, $N_{3-25}$ during the 3 noon of NPF days are 1-2 order of magnitude higher than other 20 days. Those data are within 75%-100%, but have a very high impact on mean value of $N_{3-25}$. So we also showed median values in order to give the another representative diurnal variation.

**Comment 4.** About New particle formation events: there are only three so-called NPFs during this observation period. One of them is defined by authors "Off-site NPF" during which nanoparticles were formed in PBL and transported to the site and a burst of N3-25 occurred at around 18:00, with FR at 1.64 $cm^{-3}s^{-1}$. This is contradicted. All three NPFs have different characteristics. The parameters such as FR, GR, CS may not representative for this region statistically.

**Reply and changes in manuscript.** We agree that only 3 individual events could not give representative NPF parameters for this region. Here in this study we want to give the case-based analysis for NPF events we observed, and try to find the key influencing factors. More multi-parameter and long-term studies are required to fully characterize the NPF events in Tibet Plateau and analyze the mechanisms behind them.

We define the third event as "off-site NPF" event, considering that the growing particle mode did not start from the lower limit of the detection (3 nm). Since 3-25 nm particles were not formed on site, we no longer talked about FR for this case (Line 549, Table 4).

**Comment 5.** In conclusion: Such points are not discussed in the manuscript, but in the conclusion, such as "the atmosphere of Mt. Yulong exhibited a feature of strong oxidation", "Our study provided important data in vertical profile of particles at Tibet Plateau"

**Reply and changes in manuscript.**   We apologize for the confusion. The "strong oxidation" is now changed to "stronger oxidation capacity than low attitude atmosphere", which is now described in section 3.1.1 (Line 242-244 in marked up version). "vertical profile" is now changed as "important dataset of particle physical properties" (Line 664-665 in marked up version, Line 592 in revised version).

**Comment 6.** The elevation of sampling site: 3140m or 3410m?

**Reply.** 3410m, that was a typing error and was corrected (Line 87 in marked up version).

**Comment 7.** Line 106: Silicon diffusion tube?

**Reply.** Thanks for stating that, it's now corrected in the manuscript (Line 107 in marked up version).

Response to referee comments #3 :

**Comment.**

The paper Wu et al., 2008 was cited. However, Wu et al. (2008) indicated that "Laboratory studies showed that mean diameters for the number size distributions of particles emitted by gasoline engines ranged from 40 nm to 80 nm" and "Their results showed that geometric mean diameters of particles emitted by all kinds of biofuels combustion were in the range from 110 nm to 200 nm." So the author should cited the emission test results for different kinds of fuels burning directly (diesel and gasoline traffic, biomass burning, coal, etc.) and re-write corresponding sentences.

**Reply.** Thanks for stating that. We rewrite those sentences, and cited results from emission test experiments as well as the PNSD source apportionment studies (Line 289-293 in marked up version, Line 264-270 in revised version).

[revised manuscript text omitted]